# Exploring the Biological and Physical Basis of Boron Neutron Capture Therapy (BNCT) as a Promising Treatment Frontier in Breast Cancer

**DOI:** 10.3390/cancers14123009

**Published:** 2022-06-18

**Authors:** Danushka Seneviratne, Pooja Advani, Daniel M. Trifiletti, Saranya Chumsri, Chris J. Beltran, Aaron F. Bush, Laura A. Vallow

**Affiliations:** 1Department of Radiation Oncology, Mayo Clinic Florida, Jacksonville, FL 32224, USA; seneviratne.danushka@mayo.edu (D.S.); trfiletti.daniel@mayo.edu (D.M.T.); beltran.chris@mayo.edu (C.J.B.); bush.aaron1@mayo.edu (A.F.B.); vallow.laura@mayo.edu (L.A.V.); 2Department of Hematology Oncology, Mayo Clinic Florida, Jacksonville, FL 32224, USA; chumsri.saranya@mayo.edu

**Keywords:** BNCT, boron neutron capture therapy, breast cancer, LAT-1, BPA, high-LET

## Abstract

**Simple Summary:**

BNCT is a biologically targeted, densely ionizing form of radiation therapy that allows for increased tumor cell kill, while reducing toxicity to surrounding normal tissues. Although BNCT has been investigated in the treatment of head and neck cancers and recurrent brain tumors, its applicability to breast cancer has not been previoulsy investigated. In this review we discuss the physical and biological properties of various boronated compounds, and advantages and challenges associated with the potential use of BNCT in the treatment of breast cancer.

**Abstract:**

BNCT is a high LET radiation therapy modality that allows for biologically targeted radiation delivery to tumors while reducing normal tissue impacts. Although the clinical use of BNCT has largely been limited to phase I/II trials and has primarily focused on difficult-to-treat malignancies such as recurrent head and neck cancer and recurrent gliomas, recently there has been a renewed interest in expanding the use of BNCT to other disease sites, including breast cancer. Given its high LET characteristics, its biologically targeted and tumor specific nature, as well as its potential for use in complex treatment settings including reirradiation and widespread metastatic disease, BNCT offers several unique advantages over traditional external beam radiation therapy. The two main boron compounds investigated to date in BNCT clinical trials are BSH and BPA. Of these, BPA in particular shows promise in breast cancer given that is taken up by the LAT-1 amino acid transporter that is highly overexpressed in breast cancer cells. As the efficacy of BNCT is directly dependent on the extent of boron accumulation in tumors, extensive preclinical efforts to develop novel boron delivery agents have been undertaken in recent years. Preclinical studies have shown promise in antibody linked boron compounds targeting ER/HER2 receptors, boron encapsulating liposomes, and nanoparticle-based boron delivery systems. This review aims to summarize the physical and biological basis of BNCT, the preclinical and limited clinical data available to date, and discuss its potential to be utilized for the successful treatment of various breast cancer disease states.

## 1. Introduction to BNCT

Boron neutron capture (BNCT) is an emerging radiation treatment modality that is aimed at improving tumor control while limiting damage to normal tissues. BNCT falls under the category of high linear energy transfer (high-LET) radiation, which is a form of densely ionizing radiation that causes clustered, irreparable direct DNA damage in a less oxygen dependent manner, in comparison to traditional low-LET X-ray-based therapies. Treatment with BNCT involves the targeted delivery of boronated compounds to tumor cells, followed by the irradiation of tumors with epithermal neutrons. The interaction of the nonradioactive boron-10 atoms (^10^B) with thermalized neutrons causes a nuclear capture and fission reaction, leading to the production of a high LET, low energy, alpha particle and a recoil lithium-7 atom (^10^B + ^1^n → [^11^B]* → ^4^He + ^7^Li). Given the densely ionizing nature of these high-LET particles, the biological impact of BNCT on tumor cells, compared to a reference radiation type such as X-rays, is expected to be higher. As the high LET particles deposit their energy within a radius of <10 μm, the resulting DNA damage only occurs within the diameter of a single tumor cell, largely sparing adjacent normal tissues. The dense ionization tracks produced along the DNA by high-LET radiation has been shown to produce greater DNA double strand breaks in comparison to low-LET counterparts. Additionally, studies indicate that the clustered DNA damage produced following high-LET radiation is more difficult to repair and is more likely to lead to genomic instability and cell death [1]. In addition to the above described high-LET particles, two other types of directly ionizing radiation are also produced during this reaction and contribute to the overall background radiation dose. These include γ rays formed following capture of the thermal neutrons by the hydrogen atoms in tissue, and the protons created through the scattering of fast neutrons [2,3,4]. The mechanism of action of BNCT is demonstrated in Figure 1.

The first clinical application of BNCT was performed in a patient with a malignant glioma at the Brookhaven Graphite Research Reactor. Soon after this, however, the use of BNCT in the US fell out of favor due to the occurrence of serious adverse events (including seizures, radionecrosis, somnolence syndrome), poor tumor penetration by the thermal neutrons, and the accumulation of boronated compounds in normal tissues. Despite the discontinuation of US clinical trials, BNCT continued to be investigated in Asia and Europe with the use of tumor selective boronated compounds and better penetrating epithermal neutrons beams, leading to multiple successful clinical trials. In one such trial involving malignant gliomas, the intracranial tumor bed was directly injected with a boronated compound and treated with neutron irradiation. These patients demonstrated an impressive 5-year survival rate of 58%. Similarly, Mishima and colleagues successfully treated 22 patients with cutaneous melanoma with BNCT between the years of 1987 and 2002. This study reported a complete response rate of 68.2% and a partial response rate of 23% [2,4,5].

To date, clinical use of BNCT has largely been limited to small Phase I/II trials in difficult to treat tumors or reirradiation situations. BNCT clinical trials have been carried out in a variety of disease sites, including recurrent glioblastoma, primary and recurrent head and neck cancer, sarcoma, meningioma, melanoma, hepatocellular carcinoma, and malignant mesothelioma [6,7]. A brief overview of select clinical trials is shown in Table 1.

Potential clinical advantages of BNCT over traditional external beam radiation therapy include its biologically targeted, tumor selective nature, its potentially greater clinical impact given the ability to create more irreparable double strand breaks in tumor cells in a non-oxygen dependent manner, and its capability to largely spare adjacent normal tissues while adequately treating tumor volumes. Despite these potential therapeutic advantages, challenges associated with the widespread adoption of BNCT include the limited availability of suitable neutron beams, difficulty ensuring adequate and selective boron accumulation in tumors, and our limited knowledge regarding BNCT radiation dosimetry and treatment planning aspects [8]. Due to renewed global interest in more effective cancer therapies, however, BNCT is currently being investigated and implemented at an accelerated pace. In fact, in June of 2020, two Japanese BNCT facilities started clinical BNCT therapy for unresectable, locally advanced, and recurrent carcinoma of the head and neck region, marking the first time BNCT has been reimbursed by a large health insurance system [9].

Although breast cancer is the most common malignancy in females worldwide, there has been no prospective clinical investigations regarding the feasibility of BNCT in breast cancer. To our knowledge, the only published data regarding the use of BNCT in breast cancer are case reports. In one such case report, Fujimoto et al. described the treatment of a 65-year-old woman diagnosed with a locoregional recurrence of her previously treated breast cancer within the left axilla. This disease was deemed unresectable due to the involvement of the axillary nerve and was first treated with chemotherapy and conventional radiation therapy without a durable response. BNCT was therefore administered. For the delivery of BNCT, the patient was placed in the supine position and IV boron was infused. The left axilla was then treated with neutron beams directed anteriorly and posteriorly. MRI of the axilla obtained 2 months following completion of treatment demonstrated a significant decrease in tumor mass and the reduction of pain [10]. Despite the limited clinical data to date, given its potential for achieving greater tumor cell kill, and its biologically targeted nature that allows for preferential targeting of tumor cells, BNCT offers a unique modality that may be beneficial in the treatment of a variety of breast cancer disease states, ranging from early stage to widespread metastatic disease. In this review, our aim is to provide a summary of the available preclinical data and discuss the potential therapeutic applications of BNCT in breast cancer.

## 2. Overview of Boron Carriers

An ideal boron compound for BNCT is one that preferentially accumulates in tumor, has low systemic toxicity, and is rapidly cleared from blood and normal tissues following delivery. The minimum concentration of boron-10 needed to yield lethal cellular damage following BNCT is approximately 1 × 10^9^
^10^B atoms per cell or 20 μg per each gram of tissue [11]. Since the initial discovery of BNCT, many different compounds have been manufactured. First generation boron compounds included boric acid and its derivatives synthesized in the 1950s–1960s. Given their low specificity for tumor and high systemic toxicity, these were largely discontinued from clinical use [4,11]. Second-generation boron compounds developed since then include sodium mercaptoundecahydro-closo-dodecaborate otherwise known as sodium borocaptate (BSH) and (L)-4-dihydroxy-borylphenylalanine, also known as boronophenylalanine (BPA). Since their development, these compounds have been used extensively in multiple clinical trials and have shown promising patient outcomes [11,12].

Once administered intravenously, BSH enters tumor cells via passive diffusion through the plasma membrane [13]. Studies indicate that although non-toxic, BSH has low accumulation in tumors and poor blood–brain barrier penetration [14]. Despite these issues, BSH continues to be clinically utilized in certain situations, given that it contains significantly greater amounts of ^10^B (12 times more) than BPA [13]. Hatanaka and colleagues demonstrated that in patients with gliomas, BSH pretreatment of the intracranial tumor bed followed by neutron irradiation led to a significant improvement in survival compared to historical controls [15]. Kageji and colleagues similarly administered intra-operative BNCT following intravenous administration of BSH in glioblastoma patients and demonstrated improved 2- and 5-year survivals of 31.8% and 9.1% [16]. BSH has also been successfully utilized in recurrent head and neck cancers [17]. Investigations have since been undertaken to assess whether the poor cellular uptake properties of BSH could be improved. One such attempt involved the development of a kojic-acid BSH conjugate that led to improved survival in a rat brain tumor xenograft model following BNCT administration [18]. Another attempt involved the development of poly-arginine conjugated BSH (BSH-R) that has greater cell membrane permeability in comparison traditional BSH. Interestingly, BSH-R was able to efficiently penetrate glioma and pancreatic cell lines but showed poor cell membrane permeability in breast cancer [19]. While there are no published data regarding the use of BSH in breast cancer, preclinical data available to date suggests that alternative compounds may serve as more effective boron carriers in the treatment of breast cancer.

BPA primarily enters cells through an amino acid transporter known as the L-amino acid transporter (LAT-1) [20]. BPA has been coupled to fructose to develop F-BPA to increase its solubility and improve delivery of boron to tumors [21]. Table 1 describes clinical data regarding the use of boron carriers in a variety of disease sites. The role of LAT-1 in boron transport and the efficiency of BNCT related cell killing have been demonstrated in experiments involving the transfection of LAT-1 overexpressing plasmids into T-98 glioblastoma cells. The authors of this study reported that the intracellular uptake of BPA was significantly greater in cells transfected with LAT-1 and these cells showed increased sensitivity to neutron irradiation in comparison to control cells lacking LAT-1 expression [22]. LAT-1 has been demonstrated to be overexpressed in a variety of cancer types including glioma, colorectal cancer, bladder, prostate, clear cell/renal cell, pancreatic, and breast cancer [21,23]. Although there have been no clinical studies to date using BPA as a boron carrier in breast cancer, preclinical studies assessing LAT-1 expression have demonstrated that it is overexpressed in malignant breast cancer tissues in comparison to the adjacent normal tissue counterparts. Higher LAT-1 expression has been observed in triple negative (TNBC), HER2 positive, and luminal B subtypes [24]. Additional studies have demonstrated that pre-neoadjuvant chemotherapy LAT-1 expression levels are associated with increased pathological complete response (pCR) to chemotherapy in ER negative and HER2 negative patients [25] serving as a predictive marker in this setting. In a bone metastasis model of breast cancer using xenografted breast cancer cell lines in mice tibia, BPA-F uptake and the cellular boron accumulation was found to parallel LAT-1 expression. In this model, peak intracellular boron concentrations were noted after 1 h following IV administration of BPA. In the experimental group that underwent neutron irradiation following BPA-F delivery, there was a considerable reduction of tumor growth. In contrast, the control group that received no neutron irradiation following BPA -F infusion demonstrated a continued increase in tumor size [26]. Although preliminary, this work suggests that BPA may be a good boron delivery agent in the treatment of breast cancer. It is, however, important to note that recent work suggests LAT-1 expression is downregulated in hypoxic tumors due to post-transcriptional de-stabilization of LAT-1 mRNA under low-oxygen conditions. While further evaluation of this concept is needed, smaller, non-hypoxic breast cancer lesions may be the best targets for future BPA based BNCT treatments [27]. It is important to note that recently another boronated phenylalanine derivative, commercially known as Steboronine^®^, was also approved by the Ministry of Health in Japan for clinical use. This measure came following promising phase II clinical trial data which showed Steboronine to be highly tumor selective and improve the objective response rate in recurrent and locally advanced head and neck SCC compared to historical controls [28].

In addition to these traditional boron delivery agents, there has been increased interest in developing targeted boronated therapies to increase selective tumor uptake and limit the normal tissue toxicity of BNCT. These third-generation boron delivery compounds may be divided into low molecular weight and high molecular weight agents. For instance, Boron-containing amino acids and polyhedral borane clusters are examples of low-molecular weight boron delivery agents. These compounds are of interest due to their high boron content, chemical stability, ease of incorporation into organic molecules, and selectivity for tumors [11]. Other low molecular weight compound groups currently under development include boron-containing DNA-binding molecules such as alkylating agents and polyamines, as well as receptor binding entities including as derivates of tamoxifen, estradiol, and retinoic acid [29,30]. Given that Tamoxifen is already a highly promising drug in the treatment of hormone receptor positive breast cancer, further development of Tamoxifen derived tumor-specific boron moieties may offer potential for highly effective, low toxicity BNCT treatments in breast cancer.

High molecular weight compounds such as monoclonal antibodies and other receptor-targeting moieties have also shown promising results. Nakase et al. (2020) described an antibody-based receptor targeting method using an Fc-Binding Peptide Conjugate. known as Z33. This conjugate was demonstrated to recognize and interact with the Fc domain of human IgG, making it an ideal linker to attach to receptor targeting antibodies. To deliver boron to cell, the Z33 peptide was then linked to the heavily boronated compound, dodecarbonate, and bound to the Fc component of Cetuximab. Human epidermoid carcinoma cell line A431 was treated with EGF and incubated with the Z33- DB- cetuximab compound, which led to its internalization through macropinocytosis and the intracellular release of large quantities of boron. Although this concept is still under investigation, similarly formulated antibody-conjugated boron compounds targeting HER2 or ER receptors may offer novel methods of selectively targeting breast tumors with BNCT in clinical settings [31].

Boron compounds could also be conjugated to lipids to create boron-loaded liposomes. The linkage of boron containing liposomes to monoclonal antibodies such as Cetuximab has allowed for in-vitro targeting of epidermal growth factor receptor (EGFR) expressing rat glioma cells [32]. HER2-targeted boron-containing liposomes were investigated in cell culture and showed receptor specific binding in SK-BR-2 breast cancer cells. These liposomes were noted to deliver high intracellular concentrations of boron reaching 132 ppm, and demonstrated relatively long retention times, upwards of 48 h. This work suggested that liposome-antibody conjugates may be a potent boron delivery system for the implementation of BNCT in breast cancer [33]. Liposomes may also be used to encapsulate boronated compounds. This concept was examined by Khan and colleagues following injection of mice bearing xenografted mammary tumors with liposomes carrying the heavily boronated compounds such as TAC and MAC. The tumors were then irradiated with neutrons. The authors found that boron accumulation in tumors was inversely proportional to tumor size, and smaller tumors demonstrated the best neutron treatment response. The presence of hypoxia and necrosis was hypothesized to be the cause of low boron accumulation in the larger tumors [34]. This work suggested that while liposomes are a viable method of boron delivery, this technique may be best suited for the treatment of smaller or previously untreated tumors with BNCT.

Another interesting area of research with regard to BNCT involves the development of boron cluster-modified therapeutic nucleic acids that may be utilized in gene therapy and BNCT. In one study by Kaniowski et al. (2017), antisense oligonucleotides targeting the EGFR receptor were synthesized by conjugation of DNA-oligonucleotides with a boron cluster alkyl azide component. In vitro experiments using Hela cells transfected with these constructs showed promising gene silencing and prooxidative properties leading to reactive oxygen species production. Transfected cells showed reduced cell proliferation. The authors noted that this type of dual-action compound could have binary activities of targeting EGFR overexpression in cancer and serving as a novel, targeted boron carrier for the purposes of BNCT [35].

Polymeric nanoparticles have also been investigated as potential boron delivery agents. Boron-containing nanoparticles were demonstrated to have large boron content, acceptable cellular stability, and high accumulation within tumors [11]. The most used nanoparticles in nanomedicine are gold nanoparticles (AuNPs), and preliminary studies have demonstrated that gold nanoparticles may be linked to boron to create AuNP-boron cage assemblies. Wu and colleagues demonstrated that AuNP-boron cage assemblies can be linked to anti-HER2 antibodies. Administration of the assemblies to mice bearing gastric cancer xenografts led to high levels of boron accumulation within the tumors, suggesting that this system has the potential to selectively deliver cytotoxic boron payloads necessary for BNCT treatments [36]. In an attempt to improve boron delivery in breast cancer, Li et al. (2019) created an on-demand, efficient, biodegradable boron carrier system by coating boron-nitrite nanoparticles (BNNPs) with phase-transitioned lysozymes (PTL). The BNNP-PTL particles showed high accumulation in triple negative breast cancer cell lines, and PET imaging demonstrated tumor selective uptake of the nanoparticles within breast cancer mice xenografts. Following the delivery of BNNP-PTL, the mice were injected with vitamin C, leading to on-demand in-vivo degradation of PTL, releasing the boron nanoparticles within the cellular cytoplasm. Tumor bearing mice treated with neutrons following administration of BNNP-PTL demonstrated significant tumor suppression in comparison to the control group.

The preliminary studies reviewed above indicate that a variety of compounds may serve as efficient boron carriers. Further preclinical work and in-human trials are necessary to determine the optimal boron carrier system and the appropriate concentrations necessary for effective clinical delivery of BNCT in breast cancer.

## 3. Treatment Planning Considerations and Radiobiologic Aspects of BNCT

As noted earlier, BNCT leads to the production of four different types of ionizing radiation, including γ rays protons, Li-nuclei, and most importantly, high LET alpha particles responsible for creating a majority of the BNCT related DNA-damage [2]. Deposition of energy from the high LET particles leads to the formation of high-density ionization tracks along the path of the particles, resulting in increased DNA damage compared to their low LET counterparts. This increased biological impact associated with high LET radiation is typically termed relative biological effectiveness (RBE) and is defined as the ratio of the absorbed dose from a given type of radiation to a reference type of low LET radiation (such as X-rays). Although the concept of RBE is typically used to describe the effectiveness of several types of high-LET radiation (such as carbon ions), it can only be applied with confidence when the quantity of absorbed dose can be clearly defined. In the case of BNCT, given that boron distribution may be inhomogeneous and is dependent on a myriad of factors unrelated to the quality of the neutron radiation beam, the concept of RBE alone cannot be used to estimate dose. When delivering BNCT, one must also take into account the distribution of boron in tumor and normal tissues. Therefore, a more appropriate term known as Compound Biological Effectiveness (CBE) was proposed. CBE is defined as the product of RBE and boron distribution and may be used to estimate the radiation dose delivered in tissue via BNCT. CBE is dependent on the chemical properties of the boronated compound that is being used, the route of drug administration, and the microdistribution of boron within tumor and normal tissues CBE must be determined experimentally for each compound and tissue type, however, in the clinical delivery of BNCT, assumptions based on available human and animal data are used to make estimates regarding the RBE and CBE. Ultimately, the sum of the radiation delivered to tumor and normal tissue through BNCT may be expressed as photon equivalent doses measured in Gray-equivalent (Gy-E) units and incorporates the concepts of CBE and RBE [3,37].

Currently, there are very limited data on methods of BNCT dose calculation in breast cancer. A study by Yanagie et al. (2009) described the BNCT radiation dosimetry in a phantom model of the mammary gland. Based on prior studies using BPA as the boron delivery agent, the authors estimated the boron tumor/blood ratio to be 3, tumor normal tissue ratio to be 3.5 and skin/blood ratio to be approximately 1.2. Using these ratios and irradiation using a thermal neutron beam, the authors calculated that tumoricidal radiation doses could be delivered to the mammary gland. In this model, the maximum tumor RBE dose was determined to be 42.2 Gy-Eq and the mean tumor RBE dose was estimated to be 28.9 Gy-Eq [37]. Studies in malignant brain tumors and recurrent head and neck cancers treated with BNCT have demonstrated that doses of 18–25 Gy-E delivered to at least 80% of the tumor volume are associated with improved control rates [38,39]. More recently, a newer model of BNCT dose calculation known as the photon isoeffective model was developed and validated in a limited number of studies. This model is predicted to be more accurate than the simpler RBE-CBE based method of estimating delivered dose [40]. For instance, the photon isoeffective model has been tested in head and neck cancer animal models, as well as recurrent head and neck cancer patients treated with BNCT and was found to better predict their clinical response to BNCT. Additionally, the newer model demonstrated that radiation dose to the mucosa was 30–50% higher than that predicted using the older RBE model and it was able to better predict normal tissue toxicity following BNCT [41]. This model is yet to be tested in breast tissues and provides new opportunities to more accurately estimate the dose delivered by BNCT in breast cancer.

A suitable treatment planning software is of paramount importance to accurately predict the dose distribution around the target. A well-designed BNCT treatment planning system should have a pre-processing phase with CT/MRI scans to create a geometric model, the ability to perform calculations of 3D radiation distribution, be able to estimate neutron and gamma fluences, and have a well-developed post processing phase. Several Monte Carlo-based methods were developed over the last three decades [42]. One of the earliest planning systems involving Monte Carlo based calculations was NCTPLAN developed in the 1990s which uses CT image data to create a mathematical model of the pertinent anatomic regions. This has been applied successfully in the BNCT treatment of melanoma. Another planning system that has been successfully used includes BNCT_rtpe system for treatment planning in Glioblastoma [43].

In order to clinically implement BNCT, in addition to a neutron source, a patient positioning device, robust treatment planning system (TPS), a neutron beam monitor, and a means to assess boron concentration in tumors are needed [44]. Generally, the first step is obtaining a standard CT/MRI based simulation of the patient to determine gross tumor volume, (GTV) and/or clinical target volume (CTV). During this treatment planning phase, positron-labeled BPA analogue (L-^18^F-^10^BPA) may be used to assess the in-vivo distribution of boron in-vivo. Patients may be administered L-^18^F-^10^BPA and be imaged with a PET/CT to assess the biodistribution of boron in the tumor, as well as the tumor/normal tissue ratio [45,46]. Through this, the treating physician can accurately identify patients who are most likely to benefit from BNCT based on the three-dimensional boron distribution map that demonstrates greater accumulation of boron in tumor vs. normal tissue. A PET based treatment planning system known as BDTPS was recently developed in order to integrate this macroscopic boron distribution data (obtained from the pretreatment PET scan) to better delineate targets and accurately predict the dose distribution [47]. Once treatment planning is completed, on the day of treatment, the patient is administered the boron carrier intravenously at a predetermined concentration. Following administration, the boron concentration in the blood may be assessed using plasma atomic emission spectrometry or a similar method [48]. When the boron concentration is deemed to be appropriate for radiation delivery based on predetermined thresholds, the tumor may be irradiated with neutrons.

## 4. The Potential for BNCT in Breast Cancer

### 4.1. Current Techniques in the Management of Breast Cancer

A multidisciplinary approach is applied to the management of breast cancer and typically involves breast-conserving surgery (BCS) with radiotherapy, or mastectomy with or without radiotherapy, as well as the use of neoadjuvant or adjuvant systemic therapy in select situations. Breast conserving therapy and mastectomy are both well-established treatment approaches with similar locoregional recurrence rates (LRR). When radiotherapy is delivered following a lumpectomy, a standard treatment field includes the whole breast with or without regional nodes. In the delivery of post-mastectomy radiation therapy (PMRT), the chest wall and regional nodes are usually included in the field [48]. The delivery of radiation has been shown to decrease the rate of local failure by 50% and increase the incidence of breast cancer specific survival [49,50]. Per the Early Breast Cancer Trialists’ Collaborative Group metanalysis, the delivery of PMRT improves LRR and survival in those with node positive disease, particularly if ≥4 pathologically positive nodes were detected at the time of axillary staging [51]. Additionally, studies also suggest that PMRT should be offered to patients with larger tumors The delivery of systemic therapy is influenced by the disease subtypes and the stage at presentation. In early-stage, hormone receptor positive tumors, adjuvant estrogen directed therapy is administered. In locally advanced breast cancer neoadjuvant or adjuvant chemotherapy containing both an anthracycline and a taxane are typically delivered. In those with the more aggressive triple negative subtype, following neoadjuvant chemotherapy, surgery, and radiation, further adjuvant chemotherapy with capecitabine is also recommended in attempts to improve long-term disease outcomes [49].

Below we discuss the potential for BNCT in various breast cancer disease states.

### 4.2. Inoperable Early-Stage Breast Cancer

Although BCS is well tolerated, there is a small subset of early-stage patients who refuse surgery or have extensive medical co-morbidities and thus surgery poses an excessive risk. There are currently very limited treatment options for these patients, which typically include stereotactic ablative body radiotherapy (SBRT), cryoablation, and/or systemic therapy [52,53]. Given its high-LET characteristics and the potential for increased tumor cell kill, BNCT may offer an aggressive and potentially curative treatment option in these patients through ablation of the tumor without surgery.

### 4.3. Locally Advanced Breast Cancer

In those with locally advanced breast cancer treated with upfront neoadjuvant chemotherapy, studies indicate that those who achieve a pCR have higher recurrence free survival and breast cancer specific survival (5-year recurrence free survival of 84% vs. 70% favoring the pCR group) [54]. Extensive studies conducted over the last decade have demonstrated that both locoregional and distant recurrences are heavily influenced by the tumor subtype and the molecular characteristics of the tumor. LRR and distant metastatic disease are highest in those with TNBC and lowest among those with hormone-receptor positive disease. In fact, TNBC patients who achieved a pCR were shown to have significantly improved recurrence free survival and breast cancer specific survival, while the impact of pCR on patient outcomes was much less pronounced in those with hormone receptor positive disease [49]. Given this, those patients who do not achieve a complete radiological response following neoadjuvant chemotherapy, particularly those of the triple negative variety, may be good candidates for pre-surgery BNCT as a window of opportunity trial. In contrast to external beam radiation which is more likely to result in an increased low dose to the surrounding normal tissues and organs at risk, a presurgical tumor targeted BNCT boost may improve the probability of pCR and ultimate disease control in select patient groups, while limiting normal tissue toxicity.

Additionally, in locally advanced breast cancer patients, BNCT may offer an opportunity to mobilize the immune system to improve cancer control. Khan et al. (2019) published methods describing the encapsulation of boron compounds into liposomes. They found that in murine mammary tumor models, uptake of boron containing liposomes modified the peripheral blood mononuclear cells to an antitumor phenotype, contributing to inhibition on tumor growth. This work suggests that in addition to direct tumor cell death following neutron irradiation, boron compounds themselves may be utilized to improve tumor control through immune modulation [34].

### 4.4. Locoregionally Recurrent Breast Cancer

During the treatment of malignancy with external beam radiation, inevitably some doses will be delivered to the surrounding normal tissues. As noted above, a primary advantage of BNCT lies in the fact that it is biologically targeted, thus, normal tissue damage is limited to cells with adequate boron accumulation. This principle is of particular importance in patients with locally recurrent breast cancer following prior irradiation of the chest wall. Further radiation is usually discouraged in this setting due to concerns regarding toxicity. In such locoregionally recurrent breast cancer, BNCT may offer a unique means to deliver ablative radiation doses to the site of recurrence with limited toxicity to adjacent tissues, particularly if surgical resection is not deemed to be favorable. Although it is yet to be clinically investigated, Gadan and colleagues have proposed the use of BNCT to treat locoregional recurrences in HER2+ breast cancers using immunoliposomes labeled with the monoclonal antibody, Trastuzumab. These liposomes may then act as boron carrier nanovehicles to target HER2 overexpressing cells in regions of disease recurrence [55]. A similar concept may be applied in other histological/molecular variants of breast cancer to improve disease control following the development of unresectable locoregional recurrences.

### 4.5. Metastatic Disease

In patients with oligometastatic breast cancer, BNCT may facilitate the ablation of multiple lesions in a targeted manner with curative intent. Given that BNCT allows for the biological targeting of disease based on boron distribution and is less likely to be impacted by internal organ motion, patients with oligometastatic disease within highly motion sensitive organs such as the lung and liver may particularly benefit from this therapy.

Furthermore, leveraging the proposed abscopal effect of BNCT may help improve disease control in widespread metastatic settings. Although this concept is yet to be fully investigated in clinical trials, Trivillin and colleagues described the abscopal effect of BNCT in a proof of principle experiment in xenografted colon cancer rat models. The authors found that following subcutaneous injection of BPA and irradiation of right hind leg tumors with neutrons, there was a significant reduction in left leg tumor volumes, indicating that BNCT is capable of inducing abscopal effects following treatment [56], due to its immune sensitizing properties.

BNCT may also be advantageous in the treatment of breast cancer associated metastatic brain lesions. Patients with limited intracranial disease are typically treated with radiosurgery due to its physically targeted nature, which allows for a highly favorable dose distribution, limiting damage to the normal adjacent brain. This approach, however, at times requires subjecting the patient to invasive procedures such as an immobilizing head frame, and necessitates the resource heavy process of ensuring precise targeting of radiation within brain lesions. In the setting of many brain lesions or leptomeningeal disease, whole brain radiation is often performed. Although palliative, this treatment is associated with substantial neurocognitive effects and a negative impact on quality of life due to the irradiation of significant amounts of normal brain. While novel techniques such as hippocampal sparing and medications such as memantine have improved the neurocognitive impact of whole brain radiation therapy, approaches to further reduce these side effects are being investigated [57,58]. In both of the above situations, BNCT may offer an alternative treatment option. Given that boron accumulates primarily in the tumor cells and spares normal brain cells, it will not require precise physical targeting of radiation when treating limited intracranial disease. Similarly, when treating widespread brain lesions, BNCT may have less potential for causing long term neurocognitive impacts given the biological accumulation of boron preferentially within areas of disease.

Based on data on other malignancies, BNCT has been well tolerated with minimal acute or long-term toxicity; however, infusion reactions, skin toxicity, radiation necrosis at the site of treatment, and pseudoprogression of tumors have been described in the literature [17,59,60,61]. Further studies are needed to investigate the toxicities associated with BNCT in the treatment of breast cancer.

## 5. Conclusions

Some anticipated barriers to the widespread adoption of BNCT include the need for the costly establishment of a neutron accelerator, identifying the optimal selective boron delivery agents that are most appropriate for each cancer type, and the need to develop a robust treatment planning system for BNCT delivery. These concepts must be extensively investigated in preclinical studies prior to initiating in-human clinical trials utilizing BNCT in breast cancer. Given the prevalence of BPA in clinical trials and the emerging data indicating LAT-1 overexpression in breast cancer, it is reasonable to begin any BNCT related clinical work in breast cancer using BPA as the boron carrier. Following further research, if a standardized LAT-1 expression assay can be implemented in a clinic, it may allow for the selection of high LAT-1 expressors as potential candidates for BPA based BNCT. Additionally, if novel efforts to create other boron delivery agents are successful (such as those targeted against hormone receptors and nanoparticle and liposome-based boron delivery systems), BNCT may become a possible treatment option even in breast cancer patients with intrinsically low LAT-1 expression. Studies will also be needed to compare BNCT to standard of care treatments in breast cancer patients. With further work, we believe that BNCT may be a propitious strategy for the treatment of early-stage inoperable disease, may improve pCR following neoadjuvant chemotherapy, present new opportunities for the salvage of previously irradiated locoregionally recurrent disease, and may also improve outcomes in the metastatic setting.

## Figures and Tables

**Figure 1 cancers-14-03009-f001:**
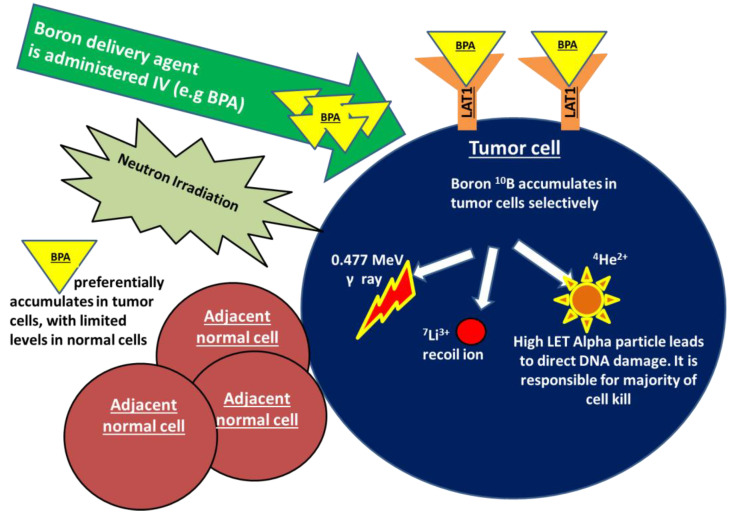
Boron neutron capture (BNCT) is a form of high linear energy transfer (high-LET) radiation. Treatment with BNCT involves the targeted delivery of boronated compounds to tumor cells, followed by irradiation of the tumor with epithermal neutrons. The interaction of the nonradioactive boron-10 atoms (^10^B) with neutrons causes a nuclear capture and fission reaction, leading to the production of a high LET alpha particle, a recoiling lithium-7, as well as γ rays and protons (^10^B + ^1^n → [^11^B]* → ^4^He + ^7^Li).The alpha particle is responsible for a majority of BNCT-induced DNA damage and cell kill. As the high LET particles deposit their energy within a radius of <10 μm, the DNA damage only occurs within the diameter of a single tumor cell, sparing any adjacent normal tissues. BPA in particular shows promise in breast cancer as a boron carrier given that is taken up by the transmembrane LAT-1 amino acid transporter that is highly overexpressed in breast cancer cells.

**Table 1 cancers-14-03009-t001:** Studies of various disease sites and outcomes with the use of BNCT. BNCT Phase I/II clinical trials have been carried out in a variety of disease sites, including recurrent glioblastoma, primary and recurrent head and neck cancer, meningioma, melanoma, hepatocellular carcinoma, and malignant mesothelioma.

**Glioblastoma Multiforme**			
**Study**	**Number of Patients**	**Boron Carrier**	**Outcomes**
Henriksson et al., 2008	30	BPA-F	Median OS: 14.2 months Median time to progression: 5.8 months
Chanana et al., 1999	38	BPA-F	Median OS: 13 months Median time to progression: 31.6 weeks
Miyatake et al., 2016	167	BPA-F	Median survival: 9.6 months
Shiba et al., 2018	7	Combination BPA with Bevacizumab	Median OS: 15.1 months Median time to progression: 5.4 months
**Head and Neck**			
**Study**	**Number of Patients**	**Boron Carrier**	**Outcomes**
Kankaaranta et al., 2012	30	BPA-F	Response rate: 76% Median PFS: 7.5 months 2 year OS: 30%
**Head and Neck (Recurrent)**			
**Study**	**Number of Patients**	**Boron Carrier**	**Outcomes**
Suzuki et al., 2014	62	BPA alone or BPA and BSH	Median survival: 10.1 months Response rate: 58% 2 year OS: 24.2%
Koivunoro et al., 2019	79	BPA-F	Complete response rate: 36% 2 year LRPFS 38% 2 year OS 21%
Wang et al., 2014, 2018, 2019	23	BPA-F	2 year locoregional control: 28% 2 year OS: 47%
Hirose et al., 2021		Steboronine^®^	Objective response rate: 71% 2 year OS for recurrent SCC: 58% Median LRPFS: 11.5 months
**Melanoma**			
**Study**	**Number of Patients**	**Boron Carrier**	**Outcomes**
Menendez et al., 2009	7	BPA-F	Overall response rate: 69% Grade 3 toxicity rate: 30%
**Meningioma**			
**Study**	**Number of Patients**	**Boron Carrier**	**Outcomes**
Takeuchi et al., 2018	31	BPA-F	Median OS: 24.6 months

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
