# Peer review of "Exploring the Biological and Physical Basis of Boron Neutron Capture Therapy (BNCT) as a Promising Treatment Frontier in Breast Cancer"

_cancers, 2022, doi:10.3390/cancers14123009_

Round 1

Reviewer 1 Report

The review written by Seneviratne et al describes the BNCT technique by giving a good introduction and some examples of its application. However, a review on breast cancer treatment is too premature for BNCT. In fact, there are not enough clinical results or applications on this topic to date. A review should report recent developments in a particular field of research, but this is not the case. For these reasons, the paper cannot be published in Cancers. 

Author Response

please find the responses in the attachment

Reviewer 2 Report

This is a well written, contributory review that summarizes the state of the art in BNCT comprehensively, taking into account the different aspects, and discusses the potential value of BNCT to treat various breast cancer disease states. It is very well presented and is not misleading in any way - it describes the potential value and constraints of BNCT in each case. It is a realistic review that I think will be very useful to readers interested in the advancement of BNCT. It is "reader-friendly". 

I have only minor suggestions:

1) Throughout the manuscript (also in Fig. 1) the authors state that BPA does not accumulate in normal tissue...this of course is "poetic license" and might be misleading to readers unfamiliar with BNCT. I think it is better to say that BPA accumulates preferentially in tumor tissue or words to that effect...

2) Page 3: non-selective accumulation of boronated compounds in normal tissues...

Should that read non-selective accumulation in tumor tissues?

3) Page 6: In addition to these traditional boron delivery agents, there have been increased interest in developing targeted boronated therapies...

Should be has instead of have...

4) Page 7: CBE is dependent on the chemical properties of the boronated compound that is being used, the route of drug administration, and the accumulation of boron in tumor vs. normal tissues.

CBE does not depend on the tumor/normal tissue boron concentration ratio...it does depend on microdistribution in tumor and in normal tissue - and of course CBE will be different in different tumor tissues and in different normal tissues, depending on different features but not on boron concentration (within a range of course). Dose will depend on boron concentration but not CBE (within a range). 

5) Page 8: This has been applied successful in the BNCT treatment of melanoma.

successfully instead of successful

6) Page 8: been successfully used include BNCT_rtpe system

includes instead of include

7) Page 8: boron concentration in the blood maybe assessed using

may be instead of maybe

8) Page 9: Per EBCTCG metaanalysis

Write out in full

9) Page 9: Although BCS is well tolerated

Write out in full

10) Page 10: immune sensitizing properties

Full stop after properties

11) Page 11: accumulation of boron only within areas of disease.

See point 1 please...replace by preferentially for example...

12) Abstract: highly overexpressed in by breast cancer cells

redundant

13) Page 3: Due renewed global interest in more effective

Due to renewed...

14) Page 8: Patients may be administered ( L-18F-10BPA and

open bracket but no close bracket

Author Response

(The authors gave the same response as above.)

Reviewer 3 Report

In this MS, the authors discuss the potential application of BNCT to breast cancer by introducing the historical background of BNCT and the state of recent clinical studies in the other types of cancer. The authors discuss the characteristics of boron compounds, such as BPA and BSH, and further mention the radiobiological aspects of BNCT. In addition, the authors discuss what kind of clinical stages or pathological states in breast cancer patient could be a candidate for BNCT. Because the cost-effectiveness is not good, and because the potential side effect of boron compounds is not slight (for example, BPA can cause severe kidney damage), I hesitate to agree that BNCT can be applied to the early stage breast cancer patients, whose prognosis is very good. However, I agree with the authors that the indication of BNCT for locally advanced breast cancer and metastatic disease with a limited neutron irradiation range should be actively considered. I am convinced that this MS will bring useful information not only to BNCT researchers but also to physicians and researchers involved in cancer treatment, since the potential of BNCT for breast cancer has been fully examined without any shortage.

Author Response

(The authors gave the same response as above.)

Reviewer 4 Report

The authors well summarized the preclinical and clinical applications of using BNCT against breast cancer in this manuscript. However, some points, listed as follows, should be modified before publication.

  1. Please correct the reaction “10B5 + 1n0 →[11B5]* → 4He2 + 7Li3” into “10B + 1n →[11B]* → 4He + 7Li” This inappropriate expression was shown in the INTRODUCTION section (page 1) and figure legend (Figure 1).
  2. In figure 1, please modify the sentence “BPA does not accumulate in normal cells” because healthy tissues do intake the BPA. It is better to say that the accumulation level of normal cells is relatively lower than that of tumors. In addition, please correct the “7Li3+” into “7Li+”.

Author Response

(The authors gave the same response as above.)

Reviewer 5 Report

The manuscript "Boron Neutron Capture Therapy: A New Treatment Frontier in Breast Cancer," submitted by Danushka Seneviratne et al, is devoted to an in-depth look at novel breast cancer therapy using radiation therapy BNCT. In this paper, the authors describe that clinical use of BNCT has been largely limited to phase I/ II trials and has focused primarily on difficult-to-treat malignancies such as recurrent head and neck cancers and recurrent gliomas. Recently, there has been renewed interest in extending BNCT therapy to the treatment of breast cancer. This review is interesting and provides important information on the physical and biological basis of BNCT, the preclinical and limited clinical data. It discusses the potential of BNCT for the successful treatment of various breast cancers with overexpressed HER2 receptor.

I recommend that this manuscript be accepted for publication after major revision.

I have some questions and comments:

  1. Boron neutron capture (BNCT) - please complete the name in the manuscript to: boron neutron capture therapy (BNCT)
  2. In the first paragraph (introduction to BNCT), the authors describe the mechanism of action of the therapy and mention the effects of radiotherapy on DNA. I suggest expanding this description and adding to the information, for example, the effect of DNA double breaks (https://doi.org/10.3389/fonc.2021.676575). This is of crucial importance in BNCT.
  3. Figure 1: My suggestion is to change the shape of the BPA from a ball shape to a triangle shape. Then this shape will fit the receptor. The spherical shape does not fit the receptor (active site) in the Y shape.
  4. Table 1: I would propose to complete table to new drug: BPA in the form of borofalan (Steboronine ®, Stella Pharma, Co. Ltd., Osaka, Japan) has been approved on March 25, 2020 for clinical use in BNCT. (doi: 10.1248/yakushi.21-00173-4)
  5. Page 6 of 14: “Boron compounds could also be conjugated to lipids to create boron-loaded liposomes. Linkage of boron containing liposomes to monoclonal antibodies such as Cetuximab has allowed for in-vitro targeting of EGFR expressing rat glioma cell”.

I propose to extend this paragraph with new conjugates of therapeutic nucleic acids anti-EGFR decorated with boron clusters in linear form and nanostructures with dual activity: anti-EGFR and BNCT radiotherapy (https://doi.org/10.3390/molecules22091393). The authors' research also focuses on breast cancer.

  1. Page 6 of 14: Please explain the abbreviations EGFR, TAC and MAC in the text.
  2. In vitro and in vivo experiments should be written in italics. Please change this in manuscript.
  3. Page 9 of 14: “breast-conserving surgery (BCT)” I would propose to change and correct the sentence to: breast-conserving surgery (BCS).
  4. I propose to remove the degrees from the authors' list of this review.

Author Response

(The authors gave the same response as above.)

Round 2

Reviewer 1 Report

The manuscript has been revised, as the authors replay that "they recognize that BNCT is not currently used or systematically evaluated in clinical trials for breast cancer" I would suggest, if possible, reevaluating the title. In fact, "Boron Neutron Capture Therapy: A New Treatment Frontier in Breast Cancer" may suggest to readers that breast cancer is already treated clinically through BNCT. For instance, "new" should be replaced with "promising", or add some information that the review focuses on the biological aspects of breast cancer... 

I hope that my suggestion will be considered by the authors before the publication.

Author Response

Dear Reviewer 1,

Thank you for your suggestion. We have adjusted the title to reflect your concerns.  

Reviewer 5 Report

Thank you very much for all comments and corrections in manuscript.

I recommend accepting this manuscript for publication.

Author Response

Dear Reviewer 5,

Thank you for your comments which helped improve our manuscript.